# Cost of childhood cancer treatment in Ethiopia

**Mizan Kiros Mirutse** [1,2]\*, **Michael Tekle Palm**[3], **Mieraf Taddesse Tolla**[1], **Solomon Tessema Memirie**[1,4], **Eden Shiferaw Kefyalew**[3], **Daniel Hailu**[5], **Ole F. Norheim**[1,6]

1 Department of Global Public Health and Primary Care, Bergen Centre for Ethics and Priority Setting (BCEPS), University of Bergen, Bergen, Norway, 2 Ministry of Health Ethiopia, Addis Ababa, Ethiopia, 3 Clinton Health Access Initiative, Addis Ababa, Ethiopia, 4 Addis Center for Ethics and Priority Setting, College of Health Sciences, Addis Ababa University, Addis Ababa, Ethiopia, 5 Department of Pediatrics and Child Health, Pediatric Hematology/Oncology Unit, College of Health Sciences, Addis Ababa University, Addis Ababa, Ethiopia, 6 Department of Global Health and Population, Harvard T.H. Chan School of Public Health, Boston, MA, United States of America

\* mizukiros@gmail.com

## Abstract

### Background

Despite the recent interest in expanding pediatric oncology units in Ethiopia, reflected in the National Childhood and Adolescent Cancer Control Plan (NCACCP), little is known about the cost of running a pediatric oncology unit and treating childhood cancers.

### Methods

We collected historical cost data and quantity of services provided for the pediatric oncology unit and all other departments in Tikur Anbessa Specialized Hospital (TASH) from 8 July 2018 to 7 July 2019, using a provider perspective and mixed (top-down and bottom-up) costing approaches. Direct costs (human resources, drugs, supplies, medical equipment) of the pediatric oncology unit, costs at other relevant clinical departments, and overhead cost share are summed up to estimate the total annual cost of running the unit. Further, unit costs were estimated at specific childhood cancer levels.

### Results

The estimated annual total cost of running a pediatric oncology unit was USD 776,060 (equivalent to USD 577 per treated child). The cost of running a pediatric oncology unit per treated child ranged from USD 469 to USD 1,085, on the scenario-based sensitivity analysis. Drugs and supplies, and human resources accounted for 33% and 27% of the total cost, respectively. Outpatient department and inpatient department shared 37% and 63% of the cost, respectively. For the pediatric oncology unit, the cost per OPD visit, cost per bed day, and cost per episode of hospital admission were USD 36.9, 39.9, and 373.3, respectively. The annual cost per treated child ranged from USD 322 to USD 1,313 for the specific childhood cancers.

from Ethiopia Health Insurance Agency. (+251) 011-557-66-98/ ehia@ethionet.et.

**Funding:** This work was supported by the University of Bergen, Trond Mohn Foundation (grant no. BFS2019TMT02), and the Norwegian Agency for Development Cooperation (NORAD) (grant no. RAF-18/0009) through the Bergen Centre for Ethics and Priority Setting (BCEPS: project number 813 596). The funders had no role in study design, data collection and analysis, decision to publish, or preparation of the manuscript.

**Competing interests:** The authors have declared that no competing interests exist.

## Conclusion

Running a pediatric oncology unit in Ethiopia is likely to be affordable. Further analysis of cost effectiveness, equity, and financial risk protection impacts of investing in childhood cancer programs could better inform the prioritization of childhood cancer control interventions in the Ethiopia Essential Health Service Package.

## Background

High-income countries have achieved remarkable progress in the survival rates of children with cancer from a low starting point (around 30%) in the 1960s to above 80% in the 2020s, even higher (90–95%) for some cancer types, such as acute lymphoblastic leukemia (ALL) and Wilms' tumor [1–5]. This achievement is mainly related to the tremendous progress made in improving access to care, timely detection, and prompt treatment; an improvement in the safety of treatment and supportive care; and a large reduction in the treatment abandonment rate [3,6,7]. On the contrary, the survival rate in low-income countries (LICs), where more than 90% of the childhood cancer burden occurs [3], is around 20–30% [3,6,8–10]. In LICs, only a small portion of children with cancer are diagnosed and treated [11]. Even if they have access to treatment, the survival rate is low due to late diagnosis (treatment initiated at an advanced stage), misdiagnosis, high abandonment rate, poor quality of care (lack of standard protocol, suboptimal trained human resources, unavailability of essential medicines, frequent stockout of drugs and supplies), high treatment-related toxicity, and poor supportive care facilities [3,11,12].

The situation in Ethiopia is similar to those of other LICs. Children with cancer are often not detected and diagnosed promptly in the country [13,14]. Those who made it through are highly likely to receive delayed, incomplete, or no care [15]. Children in the incurable disease stage (a common situation in Ethiopia) are often sent home without palliative care [13]. For many years, the country has only had one pediatric oncology unit located at Tikur Anbessa Specialized Hospital (TASH) [16] in the capital city of Addis Ababa. Recently, three additional units have opened at Jimma, Mekelle, and Gondar University Hospitals. There are critical gaps in trained human resources, the availability of diagnostic centers, and essential medicines [15]. This could generally indicate the low priority given to childhood cancer programs in Ethiopia. In 2019, the health sector recognized these critical gaps and launched a five-year strategic plan (the National Childhood and Adolescent Cancer Control Plan (NCACCP), 2019– 2023) and set a target of achieving a 40% cure rate for common and curable childhood and adolescent cancers [15]. One of the major targets set in the NCACCP to achieve this goal is increasing the number of equipped and staffed pediatric oncology units from three to eight before the end of 2023 [15]. However, the availability of evidence on the cost of running and scaling pediatric oncology units and disease (specific childhood cancer types) level cost estimates is scarce and unavailable in Ethiopia. The available estimates in LICs highly varied across study reports for several reasons, such as differences in the quality of care, cancer patterns, treatment protocols, and costing methods applied. A systematic review conducted in 2019 on the cost and cost-effectiveness of childhood cancer treatment in low—and middle-income countries (based on 18 studies that met their costing study criteria) reported an annual cost per treated child of USD 1,401 in Uganda, USD 1,638–1,913 in Rwanda, and USD 10,540 in Ghana [17]. These studies were classified as comprehensive—considering the key cost inputs assessed—and had scored > 20 out of 24 on the consolidated health economic evaluation reporting standards

(CHEERS) criteria [17]. The cost per treated child ranged from USD 2,400 to USD 31,000 in another study that examined the cost-effectiveness of treating childhood cancer in four centers in sub-Saharan African countries (Kenya, Nigeria, Tanzania, and Zimbabwe) [18].

In the recently revised Ethiopia Essential Health Service Package (EEHSP), most of the childhood cancer interventions were given low priority, as prioritization of childhood cancer interventions is mainly conducted based on experts' judgement due to the lack of local evidence on the cost and cost-effectiveness of childhood cancer interventions [19]. Therefore, the rationale behind the present costing study was to generate the evidence needed for the EEHSP and to better inform the NCACCP strategy.

## Methods

### Study setting

This study explored treatment costs at the pediatric oncology unit in TASH in Addis Ababa, Ethiopia, which was established in 2013 [16]. TASH is the largest specialized hospital in Ethiopia, with 81 clinical departments. In 2019, it had a 735 bed capacity and served close to 500,000 OPD visits (20). The pediatric oncology unit in TASH had a capacity of 42 beds and was located both within the main compound and had a satellite center around 1 km away from the hospital. The pediatric oncology unit within the TASH compound served inpatient services, while the satellite center provided both inpatient and outpatient services for regular follow-up and short cycles of chemotherapy. The pediatric oncology unit shared various services with other departments in TASH, such as pharmacy, laboratory, pathology, radiology, emergency, intensive care, and surgery. The unit was staffed with pediatric oncologists, trained oncology nurses, pharmacists, social workers in pediatric oncology services, and pediatric residents who work on a rotation basis. At the time of the study, TASH was the only hospital in Ethiopia that provided radiotherapy treatment. The pediatric oncology unit was mainly publicly funded and aimed to provide services with large subsidies, but interrupted availability of supply was a big challenge (mainly due to budget gap), and cost of care is one of the major drivers of treatment abandonment in TASH (Kiros et al.–submitted for publication). Childhood cancer treatment-related medical products were procured and distributed through the national public medical supply procuring body, Ethiopia Pharmaceutical Supply Agency (EPSA).

### Data collection and analysis

**General TASH costing study approach.** This study was conducted as part of a costing exercise carried out in TASH by the Ethiopia Health Insurance Agency (EHIA) to estimate the community health insurance premium rate (period 8 July 2018 to 7 July 2019) [20]. The additional costing elements specific to this study were collected simultaneously with data collected for the broader hospital-level costing, and the three authors of this paper had the role of guiding and coordinating the entire costing exercise (from design to analysis). The data collection was conducted October 5–28, 2020.

The annual cost of running TASH in general as well as of running the pediatric oncology department was estimated from the health system (provider) perspective using historical data. The historical costing exercise collected annual data from the period 8 July 2018 to 7 July 2019 (EFY 2011), and therefore avoided any cost distortion due to seasonal utilization differences. The exercise was structured through a mixed costing approach with predominantly top-down estimation in which aggregate costs at the hospital level were collected and allocated out to departments. This was supplemented by a bottom-up approach in estimating staff time and, to some extent, in estimating the relative consumption of certain drugs and supplies at the department level for the overall TASH costing and at the disease (childhood cancer specific)

level in the case of pediatric oncology care costing. Patient chart review was conducted to estimate the annual consumption of drugs, laboratory, radiology, and surgery supplies, and blood products for specific childhood cancers. Direct cost inputs—costs directly attributable to a specific department or service output, that is, costs of human resources, medical equipment depreciation and drugs and other supplies—were computed by estimating amounts consumed by the unit in a year (consumed quantity) multiplied by their unit costs.

*Staffing inventory*. We collected a list of all staff that was active during the study period from each department. Staff was categorized by cadre and qualification (e.g., nurse, BSc). Time allocation of each cadre of staff to each department, including patient and non-patient facing time, was collected from interviewed heads of departments for allocation. Then, for each cadre and qualification category, the total number of full-time equivalents (FTE) was calculated. This measurement considered the part-time work of staff members who shared their time across several departments. For example, if a department had two BSc nurses working 30% and one BSc nurse working 100%, the department would be recorded as having 1.6 FTE BSc nurses ($2*0.3 + 1*1 = 1.6$). Average personnel cost per cadre (including salaries, benefits, and allowances) was calculated for clinical and administrative staff employed from 8 July 2018 to 7 July 2019 based on data from the human resources and/or finance department. Staff costs were assigned based on the staff mix of the department, as defined during key informant interviews with the department head. The average annual salary plus allowances for each cadre, as defined by human resource (HR) data, were used to build up the cost of staff in each department.

**Drugs, lab reagents, and supplies** purchase costs and the volume of internal distribution of drugs, lab reagents, and supplies among departments were collected from the central pharmacy unit using the facility's Health Commodities Management Information System (HCMIS). For items where the unit cost was not found in the HCMIS, the unit cost was obtained from the EPSA. For donated items with no unit cost at EPSA, we used international unit prices such as Management Sciences for Health (MSH) [21], and The National Institute for Health and Care Excellence (NICE) [22].

*Medical equipment*. An inventory of all functional medical equipment available at the time of the visit was collected for all departments. This study included only costs related to functional clinical equipment (excluding administrative equipment such as desks, chairs, and communication equipment, for example). The value of equipment was estimated using procurement data for the study period obtained from the EPSA (3 years average), considering the equipment replacement cost regardless of whether the equipment was purchased by the facilities through EPSA, through the private sector, or was donated. A straight-line depreciation rate of 10%, which is in line with government capital item accounting standards [23], was used to amortize equipment over 10 years and to estimate the yearly cost of equipment.

*Intermediate departments and overhead services*. Shared services or departmental costs such as radiation, imaging, pathology, surgical operating room (OR), intensive care unit (ICU), pediatric emergency services (ER), inpatient food services, laundry, utilities (rent, electricity, telecommunication, water, and other utility charges) and other overhead costs (operating expenses such as office supplies, printing, educational supplies, fuel, per diem, training cost, etc.), were costed by allocating the share of each of those services used by the pediatric oncology unit using different allocation bases as appropriate in each case (for further details, see S1 Table).

*Service statistics*. Utilization data were collected from department-specific registers and service statistics reports. For cases in which this information was unavailable, we used hospital-level health management information system reports. This included total patient visits, bed days, visits by service/procedures (lab, pathology, imaging tests, surgeries), and length of stay information. The surface area of each department (in square meters) was also measured

manually. Service statistics were collected for the overall departments, as well as those specific to childhood cancer services. These service statistics were used to allocate shared costs to various departments and to compute department unit costs. For example, laundry and food were allocated to inpatient departments based on the share of total bed days; utilities, such as rent, electricity, and water, were allocated based on the square meter size of the department; other overhead costs were allocated based on the department's share of total hospital staff; and costs for administration (e.g., HR, finance, and liaison) were allocated based on the department's share of personnel and bed days, respectively. The costs of intermediate (clinical support) departments, such as the operating room, laboratory, and radiology, were allocated out to other OPD and IPD departments in the final step of the cost allocation (see S1 Fig).

We computed the total cost of the unit by adding 1) the direct costs (HR, drug, and supplies, medical equipment), 2) the share of indirect costs (food services, laundry, utilities, and other overhead cost), 3) the cost share from cross-cutting departments (such as administrative offices, liaison office), 4) the cost share from intermediate clinical support departments such as laboratory, pathology, radiology, triage, OR, pediatric emergency department, pediatric ICU, radiotherapy department) (see S1 Fig). The final cost estimate in Ethiopia Birr is converted to USD using the 2019 exchange rate [24].

To further disaggregate the costs for each specific childhood cancer, we applied different approaches. To allocate the estimated fixed costs (such as HR, medical equipment, and overhead costs) at the pediatric oncology unit level to the specific cancer types, we used the disease-level service utilization share at each department among the childhood cancer types. For intermediate clinical support departments such as laboratory, pathology, radiology, triage, ER, ICU, and surgery, we used department-specific childhood cancers' disease-specific utilization rates. Where available and reliable, each department's registry book was transcribed to find out the relative patient load for different cancer types. Accordingly, we found childhood cancer disease-specific utilization data for surgery, ER, ICU, pathology, and X-ray services. Costs for these departments were then allocated to each cancer type according to its relative share of total utilization (assuming one visit or bed day required equal resource use for all cancer types). For cases in which registry data were not available (in the case of the lab and radiology department except for X-ray), the relative consumption share among the childhood cancer types on the chart review (described below) was used as the source to determine cost distribution.

To distribute the cost of drugs and supplies from the pediatric oncology unit to specific childhood cancers, we applied the following techniques. First, with the help of a senior pediatric oncologist, we matched the drugs and supplies consumed at the pediatric oncology unit (collected from the hospital's HCMIS database) to specific disease types. This helped to identify which items belonged to which cancer types. We did one-to-one matching for medicine that was only utilized by a single cancer type, with 100% of the cost transferred to that specific cancer type. For items that were matched with two or more cancer types, we used the relative prescription rate share for that specific item (among the childhood cancer types) in the chart review to allocate the total cost of a specific drug or supply to childhood cancer types. For example, if the relative prescription rate for "x" item on the chart review was 60% acute lymphoblastic leukemia, 30% acute myeloid leukemia, and 10% Hodgkin's lymphoma, then the total cost allocated for "x" item at the pediatric oncology unit level was distributed by applying these proportions (60%, 30%, 10%, respectively).

**Patient chart review.** Since historical costing only included actual expenditures (associated with the level of quality of care) at the facility in a given period, it may not capture the full cost of treating patients. In Ethiopia, overall drug availability in public health facilities is suboptimal. The average essential drug availability estimate in 2018 was 28% [25], which might be

even lower for pediatric oncology drugs, given their high cost and low attention. The same gap in service readiness is true for lab, radiology, and pathology services. To account for such gaps, we embedded a patient's chart review in the study together with the top-down costing that considered the same study period (8 July 2018 to 7 July 2019). To obtain a representative patient's chart number for that given year, we divided the year into four quarters and selected a random month in each quarter (a total of four months selected) through an Excel-based lottery system. Then, we randomly selected a week in each selected month, a total of four weeks selected in the four quarters. This yielded 345 patients registered for the selected weeks, and all patients' charts were reviewed for a full month of clinicians' prescription order. The month was defined as 30 or 31 days, starting from the specific date they utilized care in the sampled weeks. We collected four months' prescription orders (including drugs and supplies, lab, pathology, radiology, and surgeries) and were then annualized by multiplying by 3. The annualized data were expected to be representative, given the large random sample and by assuming a similar pattern of patient flow and prescription practice in a year. The chart review was carried out October 12–23, 2020.

As indicated above, while detailed service statistics at the diagnosis level were collected for pediatric cancer departments and associated departments, data availability and quality varied and were not available at the diagnosis level in some instances. The chart review therefore afforded the secondary benefit of enabling more precise cost estimates at the disease level by providing indicative data on the distribution of services between cancer types for those departments with service statistic data gaps.

**Data collection process and data quality control.** Data collection was undertaken by experienced costing data collectors. We identified data collectors who performed well in the previously conducted costing study (in 2017) for secondary and primary hospitals by the Ethiopian Health Insurance Agency and the Clinton Health Access Initiative. The team was given a one-day training session in which the study objective, data collection tools, and guidelines and routines for data collection were covered. Data collection tools were paper-based and were derived from the Simple Cost Analysis Tool for Hospitals (SCAT) developed by Abt Associates and previously used by EHIA [20]. The team was closely managed on-site (by three of the authors in this paper), with daily check-ins at the beginning and end of each day.

Once the data collection was finalized and entered into the Excel-based tool (from the paper-based data collection templates), an iterative process of data validation was conducted. First, collected data were compiled, and preliminary analysis was performed. Gaps and suspicious values in the data were identified, and follow-up was undertaken with hospital staff in person. Follow-up was done with particularly high frequency in the weeks immediately after data collection but continued ad hoc over several months as the data were compiled and analyzed. The cost analysis was conducted using an Excel-based model adopted from the Joint Learning Network for Universal Health Coverage [26], which was previously used by EHIA for a similar exercise.

## Results

There were 42 staff members (corresponding to 32 FTE, full-time equivalent) working in the pediatric oncology unit: 3 FTE oncologists, 11 FTE residents, and 18 FTE nurses. During the study period, the unit served 1,345 patients with 7,842 OPD visits, 1,302 IPD admissions, 9.4 days of average length of stay per episode of admission, and 12,180 bed days (Table 1). Seventy percent of the bed days were in the inpatient ward within the TASH compound, and 30% were in the satellite clinic. The annual OPD visits per patient and bed days per patient were 5.8 and 9.1, respectively (Table 1).

**Table 1. Childhood cancer incident and service utilization distribution in TASH, 2018–2019.**

| Diagnosis | Share, N (%) | OPD visits, N (%) | Bed-days, N (%) | Annual OPD visits / patient | Annual bed days /patient |
|---|---|---|---|---|---|
| Acute lymphoblastic leukemia | 378 (28.1%) | 2,259 (28.8) | 2,706 (22.2) | 6.0 | 7.2 |
| Wilms' tumor | 197 (14.6%) | 967 (12.3) | 803 (6.6) | 4.9 | 4.1 |
| Hodgkin's lymphoma | 161 (12.0%) | 851(10.9) | 540 (4.4) | 5.3 | 3.4 |
| Rhabdomyosarcoma | 117 (8.7%) | 897 (11.4) | 1,366 (11.2) | 7.7 | 11.7 |
| Retinoblastoma | 90 (6.7%) | 801 (10.2) | 928 (7.6) | 8.9 | 10.4 |
| Neuroblastoma | 76 (5.7%) | 313 (4.0) | 1,081(8.9) | 4.1 | 14.2 |
| Non-Hodgkin's lymphoma | 70 (5.2%) | 400 (5.1) | 1,769 (14.5) | 5.7 | 25.5$^\Delta$ |
| Acute myeloid leukemia | 48 (3.5%) | 180 (2.3) | 1,306 (10.7) | 3.8 | 27.5$^\Delta$ |
| Osteosarcoma | 47 (3.5%) | 130 (1.7) | 412 (3.4) | 2.8 | 8.8 |
| Ewing sarcoma | 31 (2.3%) | 178 (2.3) | 507 (4.2) | 5.7 | 16.3 |
| Nasopharyngeal cancer | 27 (2.0%) | 697 (8.9) | 295 (2.4) | 25.6$^{\Delta\Delta}$ | 10.8 |
| Other cancers* | 104 (7.7%) | 169 (2.2) | 467 (3.8) | 1.2 | 3.3 |
| Total** or Average *** | 1345 (100)** | 7 842 (100)** | 12 180 (100)** | 5.8*** | 9.1*** |

\* Angiosarcoma, germ cell tumor, sacrococcygeal teratoma, yolk sack tumor, Burkitt's Lymphoma, Hemangioma, soft tissue sarcoma, Kaposi sarcoma, neuroblastoma, chronic myeloid leukemia› thymoma.

\*\* Total.

\*\*\* Average.

Δ Radiotherapy services were given as outpatient care and each day's visit was counted as a separate OPD visit. It might also be a data quality gap.

ΔΔ Can be partly explained by the high risk of treatment related prolonged neutropenia that leads to longer admission period but can also be a data quality gap.

The top seven pediatric cancer types in TASH during the study period were acute lymphoblastic leukemia, Wilms' tumor, Hodgkin's lymphoma, rhabdomyosarcoma, retinoblastoma, neuroblastoma, and non-Hodgkin's lymphoma (Table 1).

Table 2 summarizes the annual costs of running a pediatric oncology unit in TASH. The estimated annual total cost was USD 776,060 (equivalent to USD 577 per treated child). Seventy-eight percent of total costs consisted of direct and indirect (overhead) costs from the pediatric cancer OPD and IPD departments (Table 2). From this share, drugs and supplies made up the major component (33%), followed by personnel costs (27%). The remaining 22% of total costs were evenly split between other clinical department services (ER, ICU, and surgery), which represented 11% of total costs, and intermediate department services (lab, pathology, radiology, and triage), which represented 11% of total costs. Among intermediate departments, radiology services accounted for the largest cost, while the ER accounted for the largest cost share among the clinical departments.

Thirty-seven percent (USD 289,953) of the total cost was attributable to OPD services, and the remaining 63% (USD 486,108) was attributable to IPD services. For the pediatric oncology unit, the cost per OPD visit, cost per bed day, and cost per episode of hospital admission was USD 36.9, 39.9, and 373.3, respectively.

The cost of running a pediatric oncology unit per treated child ranged from USD 469 to USD 1,085 on the scenario-based sensitivity analysis (S2 Table), which accounted for the potential cost underestimation due to under-provision of services, using the cost estimation results from the patients' chart review.

Table 3 presents costs for the most prevalent cancer types per patient, per OPD visit and per bed day. Overall, the annual cost per patient ranged from USD 322 to USD 1,313, but the estimate for the top six cancer types was in the range of USD 433 to USD 676.

Table 4 presents cost drivers for the different cancer types. Drugs and supplies were the largest cost contributors to pediatric cancer treatment at TASH. Acute lymphoblastic leukemia, Wilms' tumor and Hodgkin's lymphoma had a cost share of drugs and supplies that was

**Table 2. Annual costs of treating childhood cancers in TASH July 2018–July 2019.**

| Pediatric oncology OPD (including radiotherapy) and IPD | Annual total cost, USD (%) | Annual cost/patient (USD) |
|---|---|---|
| Personnel | 212,367 (27) | 157.9 |
| Drugs & Supplies | 258,391 (33) | 192.1 |
| Equipment depreciation | 11,649 (2) | 8.7 |
| Overhead | 121,642 (16) | 90.4 |
| **Intermediate** | | |
| Lab | 21,112 (3) | 15.7 |
| Pathology | 14,231 (2) | 10.6 |
| Radiology | 43,885 (5) | 32.6 |
| Triage | 5,733 (1) | 4.3 |
| **Other clinical departments** | | |
| Pediatric ER | 65,875 (8) | 49.0 |
| Pediatric ICU | 15,509 (2) | 11.5 |
| Pediatric Surgery | 5,667 (1) | 4.2 |
| Total | 766 060 (100) | 577.0 |
| Distribution by departments | | |
| Department | Cost (%) | Cost per service utilization |
| Out-Patient Department (OPD) | 289,953 (37%) | USD 36.9 per OPD visit |
| In-Patient Department (IPD) | 486,108 (63%) | USD 39.9 per bed day |
| | | USD 373.3 per episode of admission |

above the pediatric oncology unit level estimate (33%). However, non-Hodgkin's lymphoma, acute myeloid leukemia and nasopharyngeal cancer had low drugs and supplies and high HR share.

## Discussion and conclusions

The total annual cost of running a pediatric oncology unit in TASH was USD 776,060. The cost increased to USD 1,122,802 (a 45% increase from the base case scenario) when the potential under-provision of services due to interrupted availability of drugs and supplies, lab,

**Table 3. Disease-level childhood cancers unit costs in TASH July 2018–July 2019.**

| Pediatric cancer diagnosis | Total cost | Cost share (%) | Annual cost per patient (USD) |
|---|---|---|---|
| Acute lymphoblastic leukemia | 219 481 | 28.3 | 581 |
| Wilms' tumor | 90 383 | 11.6 | 459 |
| Hodgkin's lymphoma | 69 733 | 9.0 | 433 |
| Rhabdomyosarcoma | 67 174 | 8.7 | 575 |
| Retinoblastoma | 47 960 | 6.2 | 535 |
| Neuroblastoma | 51 357 | 6.6 | 676 |
| Non-Hodgkin's lymphoma | 65 778 | 8.5 | 940 |
| Acute myeloid leukemia | 62 443 | 8.0 | 1313 |
| Osteosarcoma | 18 510 | 2.4 | 396 |
| Ewing sarcoma | 20 084 | 2.6 | 644 |
| Nasopharyngeal cancer | 32 390 | 4.2 | 1 187 |
| Other cancers | 30 766 | 4 | 322 |

**Table 4. Disease-level cost drivers for childhood cancers in TASH, 2018–2019.**

| Pediatric cancer diagnosis | Cost category share (in %) | | | | | | |
|---|---|---|---|---|---|---|---|
| | Human resources | Drugs & supplies | Equipment | Overhead | Clinical Support | Intermediate | Total |
| Acute lymphoblastic leukemia | 24 | 45 | 1 | 13 | 12 | 5 | 100 |
| Wilms' tumor | 20 | 39 | 1 | 11 | 11 | 18 | 100 |
| Hodgkin's lymphoma | 21 | 37 | 1 | 10 | 16 | 14 | 100 |
| Rhabdomyosarcoma | 30 | 29 | 3 | 20 | 9 | 10 | 100 |
| Retinoblastoma | 33 | 28 | 5 | 20 | 5* | 9 | 100 |
| Neuroblastoma | 30 | 24 | 2 | 20 | 13 | 12 | 100 |
| Non-Hodgkin's lymphoma | 42 | 18 | 1 | 25 | 10 | 5 | 100 |
| Acute myeloid leukemia | 32 | 12 | 1 | 19 | 31 | 5 | 100 |
| Osteosarcoma | 29 | 40 | 2 | 20 | 0* | 9 | 100 |
| Ewing sarcoma | 32 | 24 | 3 | 23 | 9 | 9 | 100 |
| Nasopharyngeal cancer | 43 | 25 | 6 | 15 | 3* | 9 | 100 |
| Other cancers | 19 | 43 | 1 | 12 | 10 | 15 | 100 |

Clinical Support includes ER, ICU, surgery.

Intermediate includes X-ray, CT scan, ultrasound, pathology, lab, MRI, ECO.

* A zero or a small number of patients were reported from the clinical support departments.

pathology, and radiological investigation were accounted for using chart review. The annual cost of running a pediatric oncology unit per patient (for the base case scenario) was USD 577 and ranged from USD 469 to USD 1,085 (S2 Table) on the scenario-based sensitivity analysis. The major drivers (78%) of the costs were drugs, supplies, and personnel. The true cost of running a pediatric oncology unit could be higher than our estimate if we included the start-up investment costs (such as building and training costs), and it could even increase further as the availability of quality care (advanced diagnostic services, continuous availability of primary and supportive treatment, palliative care) improves.

Overall, the cost composition of the pediatric oncology unit was comparable to the adult oncology unit but markedly differed from the hospital-level (TASH) cost [20]. HR was the main cost driver at TASH, whereas drugs and supplies took the largest share at the pediatric and adult oncology unit level (S3 Table). This can mainly be explained by the heavy dependence of oncology units on expensive chemotherapies. This pattern is also similar for the common childhood cancer types (Table 4) but particularly marked in the case of ALL, and Wilms' tumor which can be partly explained by the longer duration of chemotherapy, the high number of chemotherapy drugs used per cycle in the case of ALL, and the relatively expensive drug types (treatment protocol) in the case of Wilms' tumor. Non-Hodgkin's lymphoma, acute myeloid leukemia and nasopharyngeal cancer had low drugs and high HR cost share, and this could be related to a smaller number of drugs needed per cycle in the case of AML and nasopharyngeal cancer.

For the pediatric oncology unit, the cost per OPD visit, cost per bed day, and cost per episode of hospital admission were USD 36.9, 39.9, and 373.3, respectively. The cost per pediatric cancer unit OPD visit, excluding clinical support departments such as the ER, was USD 27. This is significantly higher than the hospital-level average of USD 15.7 [20]. One explanation is that expensive chemotherapy treatment is commonly administered as outpatient care. A second explanation is that radiotherapy treatment has been included in this estimate (since radiotherapy was delivered as outpatient care for children), which is a more advanced form of OPD treatment than the average visit.

The cost per pediatric cancer unit IPD bed day, again excluding clinical support departments such as ICU, was USD 37.3. This is significantly lower than the hospital average of USD 61.3 [20]. At the hospital level, nearly one-third of IPD costs came from intermediate departments, whereas the cost share of intermediate departments for pediatric cancer treatment was found to be around 11%. This discrepancy may be one explanation for the difference in the cost of IPD per bed day. The second reason is the higher service utilization in the pediatric oncology unit (12,180 bed days), lowering the cost per bed day estimate. This could be an economy of scale but could also be related to suboptimal provision of services due to an imbalance between patient flow (volume of services needed) and facility preparedness.

For specific childhood cancer types, the annual cost per patient ranged from USD 322 to USD 1,313, but the estimate for the top six cancer types was in the range of USD 433 to USD 676. The less commonly reported childhood cancer types had higher annual unit cost estimates, which can partly be explained by the low volume of service utilization that exaggerated the unit cost estimate or could be related to the data quality gap.

Generally, it was difficult to perform a one-to-one comparison of our study with reports from similar settings, mainly due to the inconsistency in costing approach and methods, differing cancer patterns, treatment protocols, and quality of care. Our estimate is smaller compared to study reports from Uganda and Rwanda, which applied a relatively similar costing approach [27,28]. We adjusted the estimate from both countries (which was in 2014) to match our study period (2018–2019) using the inflation rate [29,30]. In Uganda, the annual cost of treating a child with Burkitt's lymphoma in 2019 was US$1,479 (28). In Rwanda (in 2019), the cost per treated child for Hodgkin's lymphoma and Wilms' tumor was USD 1,757 and USD 1,345, respectively [27]. Beyond service quality differences, differing disease patterns (heterogeneity), and treatment protocols, costing methods explain part of the variation in the unit cost estimate. The cost of providing health services in Ethiopia is generally low since the HR salary wage is low, and most utilities (such as water and electricity) are subsidized by the government. For example, the average doctor's salary per month in 2022 is USD 408 in Ethiopia, USD 1,600 in Rwanda, and USD 1,740 in Uganda [31–33]. Similarly, the cost of electricity per kWh is USD 0.007, 0.19, and 0.25 in Ethiopia, Uganda, and Rwanda, respectively [34].

Our cost estimate could help the government and other stakeholders in Ethiopia make informed investment decisions. For an annual cost of USD 577 (which could be as high as 1,085 when adjusted for suboptimal care) per treated child, the budget impact of financing the childhood cancer program—combined with prioritizing high-impact interventions—could be low as the population in need of care is small (annual incidence of childhood cancer is around 3,800) [35], hence it could be affordable in Ethiopia. Of course, the cost of running a pediatric oncology unit and the cost per treated child could increase as more centers, beds, trained staff, advanced diagnostics, and safer treatment are made available, which could also significantly improve the survival rate of children with cancer. Cost must also be compared to expected health outcomes (but no cost-effectiveness analysis has been conducted to date in Ethiopia). There is growing evidence that investing in childhood cancer programs could be cost-effective in LMICs [6,17,18,36,37] and the incremental cost-effectiveness ratio estimate ranged from USD 22 to USD 4,475 per DALY averted [17]. One of the key pillars of universal health coverage is addressing equity (through prioritizing the worst off) and financial risk protection [38]. Children with cancer fall under the worst-off categories as they face a large individual-level disease burden, and their guardians are at high risk of encountering impoverishing health expenditure, as the cost of cancer care (direct and indirect medical and indirect non-medical costs) is high [38–41]. The high share of drugs and supplies costs in our study could also indicate the potential cost burden to affected households, if not made continuously available with a large subsidy or cost exemption through public funding. This is critical given that around 23.5% of

Ethiopians live below the poverty line [42]; the nation's health insurance system is in its infancy [43,44] and does not provide effective coverage for specialty services such as childhood cancer treatment.

Our study has many limitations that warrant cautious interpretation of its results. First, key costing elements, such as building costs and long-term specialty training to produce pediatric oncologists, oncology nurses, pharmacists, and pathologists, were not captured. Second, despite the effort we made to address the underestimation of costs due to suboptimal provision of services, the prescription practices of clinicians could be influenced by the perceived availability of services in TASH. As a result, clinicians might prescribe less effective (less costly) alternatives or avoid prescribing such items to patients. Further, poor chart documentation of the clinician's order might affect our cost estimate. Third, non-medical costs, such as transport, lodging, and productivity loss of guardian's, were not included. Fourth, despite the rigorous efforts made to improve the data quality (related to the poor hospital recording), there is a chance that our results could be biased, most likely in the direction of underestimating cost.

In conclusion, running a pediatric oncology unit in Ethiopia is likely to be affordable. Further analysis of the cost effectiveness, equity, and financial risk protection impacts of investing in childhood cancer could strengthen our findings and better inform the national essential health service package. An integrated and robust health, human resource, supply, lab, and financial information management system, and a cancer incidence and survival registry are highly needed for a better cost and health outcomes estimate and monitoring of progress and, more importantly, to enhance day-to-day decisions in TASH and the pediatric oncology unit.

## Supporting information

**S1 Fig. Cost aggregation at pediatric oncology unit level in TASH July 2018–July 2019.**
(PDF)

**S1 Table. Allocation statistics used for costing analysis.**
(DOCX)

**S2 Table. Scenario-based cost sensitivity analysis in TASH July 2018- July 2019.**
(DOCX)

**S3 Table. Cost categories share for TASH overall, adult oncology and pediatric oncology units in TASH, 2018–2019.**
(DOCX)

## Acknowledgments

We would like to thank Ethiopia Health Insurance Agency and Tikur Anbessa Specialized Hospital for their leadership and administrative support.

Ethical approval

We obtained ethical approval for the study from the Regional Committee for Medical and Health Research Ethics (REC Western Norway, approval no. 64245), and data use approval for the fully anonymized cost dataset from the Ethiopia Health Insurance Agency (አጤሞኤ/ሰጉ./ 999/014)

## Author Contributions

**Conceptualization:** Mizan Kiros Mirutse, Michael Tekle Palm, Mieraf Taddesse Tolla, Solomon Tessema Memirie, Ole F. Norheim.

**Data curation:** Mizan Kiros Mirutse.

**Formal analysis:** Mizan Kiros Mirutse, Michael Tekle Palm.

**Funding acquisition:** Ole F. Norheim.

**Investigation:** Mizan Kiros Mirutse, Michael Tekle Palm, Daniel Hailu.

**Methodology:** Mizan Kiros Mirutse, Michael Tekle Palm, Mieraf Taddesse Tolla, Solomon Tessema Memirie, Eden Shiferaw Kefyalew, Ole F. Norheim.

**Supervision:** Mieraf Taddesse Tolla, Solomon Tessema Memirie, Ole F. Norheim.

**Writing – original draft:** Mizan Kiros Mirutse.

**Writing – review & editing:** Mizan Kiros Mirutse, Michael Tekle Palm, Mieraf Taddesse Tolla, Solomon Tessema Memirie, Eden Shiferaw Kefyalew, Daniel Hailu, Ole F. Norheim.

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
