## [Decision Letter · Decision Letter 0]

17 May 2023

Cost of childhood cancer treatment in Ethiopia

PONE-D-22-25209

Dear Dr. Mirutse,

We’re pleased to inform you that your manuscript has been judged scientifically suitable for publication and will be formally accepted for publication once it meets all outstanding technical requirements.

Kind regards,

Muktar Beshir Ahmed, MPH

Academic Editor

PLOS ONE

1. Please provide additional details regarding participant consent. In the ethics statement in the Methods and online submission information, please ensure that you have specified (1) whether consent was informed and (2) what type you obtained (for instance, written or verbal, and if verbal, how it was documented and witnessed). If your study included minors, state whether you obtained consent from parents or guardians. If the need for consent was waived by the ethics committee, please include this information.

Reviewers' comments:

Reviewer's Responses to Questions

**Comments to the Author**

1. Is the manuscript technically sound, and do the data support the conclusions?

Reviewer #1: Yes

Reviewer #2: Yes

2. Has the statistical analysis been performed appropriately and rigorously? 

Reviewer #1: Yes

Reviewer #2: Yes

3. Have the authors made all data underlying the findings in their manuscript fully available?

Reviewer #1: Yes

Reviewer #2: Yes

4. Is the manuscript presented in an intelligible fashion and written in standard English?

Reviewer #1: Yes

Reviewer #2: Yes

5. Review Comments to the Author

Reviewer #1: Really nice work.this study shows the importance of resource stratification in LMIC and we need to develop more research related childhood tumors. There should be more studies and literature review showing the situation in other African countries

Reviewer #2: Dear Author,

Thank you for this piece of work that was done so well. I have read through your work and have found it informative and well-written. I have nothing to add to it and congratulations upon fine writing.

6. PLOS authors have the option to publish the peer review history of their article (what does this mean?). If published, this will include your full peer review and any attached files.

Reviewer #1: **Yes: **Shah Zeb Khan, MD

Reviewer #2: No

---

## [Editor Report · Acceptance letter]

24 May 2023

PONE-D-22-25209 

Cost of childhood cancer treatment in Ethiopia 

Dear Dr. Mirutse:

I'm pleased to inform you that your manuscript has been deemed suitable for publication in PLOS ONE. Congratulations! Your manuscript is now with our production department. 

Kind regards, 

on behalf of

Professor. Muktar Beshir Ahmed 

Academic Editor

PLOS ONE